## [Transparent Peer Review file · Nature Communications]

Single-cell multi-omics resolved analysis of mitochondrial genome-wide mutational burden constraint and mosaicism

Corresponding Author: Professor Leif Ludwig

Version 0:

Reviewer comments:

Reviewer #1

(Remarks to the Author)

This is an interesting paper which presents a very thorough bioinformatics analysis of cell lines with an inducible mutation in POLG (KI) compared to control lines, followed by an analysis of human patient blood cells from individuals with known pathogenic mtDNA mutations. The main novel finding is a much greater number of mtSNVs in the POLG KI cells than anticipated, and an increase in mtSNV rate in the human blood cells. This is supplemented by a detailed annotation of the mtSNVs which show a pattern consistent with selection against functional variants (which is expected and confirmatory). The technical approach is cutting edge and the analysis thorough, with a very sensible novel development of variant constraint analysis which encompasses heteroplasmy.

- It is not always explicit whether the analysis is based on data at single cell resolution or in pseudobulk. It would help to show the cell-level mtDNA coverage. I infer this to be very in controls low because the 2-3 fold increase described in POLG KI mutant cells was <30x, so does this mean the control coverage was <<10x? The current draft presents normalised coverage which conceals this raw data. This is very important because low coverage will drastically limit the ability to detect and measure heteroplasmy levels accurately – introducing a systematic bias between the cell lines which will not be addressed in a pseudobulk analysis. I am afraid this is potentially a fundamental flaw.
- The POLG KI line is inducible allowing the authors to turn on and turn off the mutator phenotype. This is very elegant but not capitalised on. Why didn't the authors compare the same cell line before and after a burst of mutagenesis? This would control for many of the confounders they struggle with (eg inherited variants – and critically, the method used to generate the cell line itself which could induce mutations, or lead to rapid segregation of pre-existing mutations through bottleneck effects). I am afraid this could also be another fundamental flaw influencing the main result.
- The hash-tagging method is not without difficulties but the authors do not show that their method reliably separates the different cell lines. This is critically important. We need to be confident that the authors are able to reliably resolve single cell sequences to measure the mutation burden in single cells.
- Orthogonal validation. The methods they are using are novel and it would be reassuring to see some orthogonal validation eg of the differences in mtDNA content (related to depth), and the presence of linearised mtDNA at high levels. Genuine bulk analysis of the mtDNA, deletion and transcriptomes would provide this.
- It would be extremely reassuring to see the effects of growing the cell lines in galactose, which forces oxidative metabolism. I would anticipate that the selection signal would be much stronger – validating their conclusions experimentally, rather than by inference from single observations in each cell line.
- What is the provenance of the cell lines? Eg number of passages for the primary and POLG KI lines?
- I am unclear whether their pipeline excludes the confounding effects of mtDNA sequences embedded in the nucleus (important for the low-heteroplasmy variants)
- Definition of de novo variants. As the authors note, there is a threshold for detection of mtSNVs linked to mtDNA coverage. It is therefore impossible to be sure whether any variant is strictly 'de novo' or whether it was present in a small number of cells at low heteroplasmy levels. The authors must temper their conclusions/results with this in mind.
- Related to this – the ability to detect low level heteroplasmies is dependent on the sequencing depth. Their own data shows site-specific differences in coverage. Did the authors account for this in their analysis? Particularly when comparing different mutation classes (eg could it be that pathogenic sites tend to have low coverage, thus limiting their potential for detection). It would be useful for the field to show whether this is an important confounder or not.

- The authors have previously used mtDNA to lineage trace, but they do not do this in the current paper, despite their assertion that there are sub clones within their cell lines. I think this needs greater exploration before they reach the conclusions that they do.
- Finally, the manuscript refers to phenotypes on several occasions, but their current analysis does not systematically study phenotype. Eg 'these findings clearly suggest POLGD274A phenotypes to be driven by the cumulative mutational burden as compared to the impact of individual or a few deleterious variants as observed in classical mitochondrialopathies.' I am afraid this conclusion cannot be drawn from the data presented which does not include these phenotypes and certainly doesn't show a causal role.

(Remarks on code availability)

Reviewer #2

(Remarks to the Author)

In this manuscript by Hsieh et al, the authors leverage a technique mitochondrial single-cell ATAC-seq, which they previously established to investigate heteroplasmy levels of specific variants from patients with mtDNA disease, (Nature Genetics 2022) to measure mtDNA mutational loads and somatic mosaicism. To bench-mark their technique they use HEK-293 cell lines with a knock-in mutation in POLG which causes a proof-reading deficiency of the mtDNA polymerase and elevated mtDNA mutation rates. Interestingly in these lines, the variant allele frequencies (VAFs) of individual cells did not reach classical thresholds for metabolic dysfunction. However, when complex I and complex IV activity was measured in bulk cell cultures, there was metabolic impairment, suggesting multiple mutations can act synergistically to affect mitochondrial function.

The authors then introduce two metrics: single-cell mtDNA mutations per million base pairs (scmtMPM) and heteroplasmy-weighted mitochondrial local constraint scores (scwMSS). These useful measures allow comparative analysis of samples from subjects of different ages and disease groups which is very useful for the research community. They then apply their benchmarked technique and measures to test datasets including control blood samples from young (5 year old) and older (47 year old) controls and 4 patients with the m.3243A>G mutation (aged 29-80). They show that the somatic mtDNA mutation load per cell increases with age in control samples and suggest that the m.3243A>G mutation may affect the accumulation of somatic mtDNA mutations.

Specific comments:

1. The authors have established a protocol to quantify mtDNA mutation load at the single cell level at an unprecedented resolution. Previous single cell work on somatic mtDNA mutations has been at the level of 100s of cells, this was 1000s. The data is very high resolution and supports previous studies which have shown similar mutational spectra and dynamics of age-related somatic mtDNA mutation accumulation.
2. This work is of high technical quality and provides a robust framework for assessing mtDNA load/constraints/evolution in other tissue/disease subgroups.
3. I think the title could be clearer, as although the technique is technically multi-omic, other than one experiment towards the end where the nuclear ATACSeq data is used to subset immune-cells, this paper is really focussed on mtDNA sequence analysis. There is very little discussion of the impact (if there is any) of cell type on somatic mtDNA mutagenesis. The majority of the analysis groups all of the different cell types together. I would suggest either fully describing all of the data at the level of cell types, or just using the name of the technique in the title (mitochondrial single-cell ATAC-seq) which will show the focus of the paper.
4. There appears to be three distinct populations of cells in the K136 cell line, but the data is analysed as one population. Are there specific clonal variants that are driving the clustering in this line?
5. Can you be more explicit when describing the confirmed pathogenic mutations on Mitomap? There are a number of different tables on there and I'm guessing you used the 'confirmed' table. It would be good to state this to avoid confusion.
6. For the scmtMPM score, from what I understand, this is normalised to coverage? Can you be explicit that this is the total of the mtDNA aligned reads and not the overall sequencing depth of the ATAC-Seq?
7. I don't really understand how the conclusion was drawn that the 3243 mutation impedes somatic mtDNA mutation accumulation? In figures 5d and 5e both of these patients are relatively young (29 and 35) and the total scmtMPM (-3243A>G) looks to be slightly lower than H47 (closest age-match). However, M60 and M80 are much higher supporting the accumulation of somatic mtDNA mutations with age. What is the evidence that the 3243 variant impedes the accumulation of somatic variants? What happens if you draw similar plots for M60 and M80? I may be misunderstanding the figure, is there an alternative way of presenting this data?

(Remarks on code availability)

This is not my area of expertise so I did not attempt to run the code.

Reviewer #4

(Remarks to the Author)

The study by Yu-Hsin Hsieh and colleagues presents a study in mitochondrial genomics through the development and application of novel single-cell metrics to quantify mitochondrial DNA (mtDNA) mutational burden and heteroplasmy dynamics. Utilizing mitochondrial single-cell ATAC-seq (mtscATAC-seq), the authors introduce two key metrics—single-cell mtDNA mutations per million base pairs (scmtMPM) and heteroplasmy-weighted mitochondrial local constraint scores (scwMSS)—to capture genome-wide mtDNA variation at single-cell resolution. The study benchmarks these metrics in POLGD274A knock-in cell lines and applies them to primary human immune cells from both healthy donors and patients with mitochondrial encephalomyopathy (MELAS). While the work addresses an important challenge in mitochondrial genetics, offering potentially valuable insights into mitochondrial disease mechanisms, several critical issues and concerns warrant further attention.

Major Comments:

1. The authors reported 455 variants in control cells versus 2484 and 4651 variants in the KI36 and KIA2 POLGD274A knock-in cells, respectively. Given the substantially higher sequencing depth in the knock-in lines, the increased variant counts may largely reflect depth-related bias. Additionally, the authors should also clarify how uneven mtDNA coverage across the mitochondrial genome was accounted for during variant calling and downstream analyses.
2. The two metrics (scmtMPM and scwMSS) represent key contributions of this study. However, biases in variant detection and variability in sequencing coverage could substantially influence their accuracy. The reported median mtDNA sequencing depth, ranging from approximately 10x to 40x, is very low for confident variant calling, particularly for low-frequency variants. Although the authors have implemented measures to minimize these biases, further validation, such as targeted deep sequencing, is needed to confirm the accuracy of variant calls and to robustly support the conclusions drawn from these metrics.
3. The analysis assumes that variants unique to POLGD274A lines are de novo, but clonal expansions or bottlenecks could also influence variant distributions. Could the authors elaborate on how these factors were considered or controlled for in their analyses?
4. The authors incorporate a mitochondrial genome constraint model based on population genetics data from gnomAD. Have the authors considered the impact of population diversity, mitochondrial haplotypes, as well as the cell-type-specific or context-dependent functional constraints?
5. Have the authors explored integrating other variant annotations, like predicted pathogenicity scores or conservation, to help refine the MLC-based scoring?
6. Given that mtscATAC-seq simultaneously captures mtDNA variants and chromatin accessibility, have the authors investigated potential interactions between mtDNA mutational burden and nuclear chromatin states? Such as, whether/how nuclear-mitochondrial interactions influencing variant dynamics? whether cells with higher mutational burden exhibit altered chromatin accessibility; Additionally, single-cell Multiome technique, which profile both transcriptomes and chromatin accessibility in individual cells, could provide an even more powerful platform to investigate the functional consequences of mtDNA mutations. Have the authors considered applying or integrating such approaches?
7. It would strengthen the impact of the study if the authors included functional assays to see how mtDNA variants actually affect cells. Further linking the scmtMPM and scwMSS metrics to cellular phenotypes would add important biological context to the current study.
9. Could the authors expand on the translational implications of their findings. In particular, discussing how feasible it might be to apply the scmtMPM and scwMSS metrics in clinical or therapeutic settings, along with any potential challenges, would add valuable perspective.
9. The manuscript currently lacks enough detail about the statistical analyses used. The authors should include clear descriptions of the statistical methods applied throughout the study to better support their findings and improve transparency.

Other Comments:

It would be helpful if the authors provided a summary table listing all variants detected in this study, including average sequencing depth, the number of cells carrying each mutation within each group, and other relevant information.

(Remarks on code availability)

Version 1:

Reviewer comments:

Reviewer #1

(Remarks to the Author)

I thank the authors for their thorough and meticulous response to my first review, and particularly for the generation of new data. This, of course, raises further questions which they may want to consider.

1. The authors generated more reads from the three cell lines. Does the increased read depth influence their results and conclusions?
2. Why is the read depth different between the cell lines? I presume this reflects a difference in mtDNA content. Has this been validated using an orthogonal method?
3. The comparison of heteroplasmy value using pseudo and real-bulk analysis really misses the point I was making – that low coverage compromises the ability to call and measure heteroplasmy reliably. I do not think the current data really addresses this.
4. The authors should reword their section on the 'metabolic bottleneck'. As written, it conflates two issues: (1) a bottleneck – if present you should see heteroplasmy levels changing by accelerated random drift – or the random loss/gain of heteroplasmies; (2) the metabolic stress which they suggest has killed a large proportion of cells. If this is the case, then you would expect less pathogenic variants after the metabolic challenge. It is not clear to me whether the authors saw this or not.

(Remarks on code availability)

Reviewer #2

(Remarks to the Author)

I enjoyed reading the revised version of this manuscript. I am happy that that authors have addressed my comments and the additional analyses have strengthened the manuscript.

(Remarks on code availability)

Reviewer #3

(Remarks to the Author)

I thank the authors for their thorough responses. My previous concerns have been addressed, and I have no further comments on the manuscript.

(Remarks on code availability)

Response to Reviewers

We thank the Reviewers for the very thorough review of our work, which helped to significantly improve our manuscript. We provide a detailed point-by-point response to all comments in blue and have modified the manuscript based on the valuable suggestions of the Reviewers. Corresponding changes in the manuscript are also highlighted in blue.

Reviewer #1 (Remarks to the Author):

This is an interesting paper which presents a very thorough bioinformatics analysis of cell lines with an inducible mutation in POLG (KI) compared to control lines, followed by an analysis of human patient blood cells from individuals with known pathogenic mtDNA mutations. The main novel finding is a much greater number of mtSNVs in the POLG KI cells than anticipated, and an increase in mtSNV rate in the human blood cells. This is supplemented by a detailed annotation of the mtSNVs which show a pattern consistent with selection against functional variants (which is expected and confirmatory). The technical approach is cutting edge and the analysis thorough, with a very sensible novel development of variant constraint analysis which encompasses heteroplasmy.

We thank the Reviewer for their praise and for acknowledging the advances presented in our manuscript.

1. It is not always explicit whether the analysis is based on data at single cell resolution or in pseudobulk. It would help to show the cell-level mtDNA coverage. I infer this to be very low in controls because the 2-3 fold increase described in POLG KI mutant cells was $<30\times$, so does this mean the control coverage was $\ll 10\times$? The current draft presents normalised coverage which conceals this raw data. This is very important because low coverage will drastically limit the ability to detect and measure heteroplasmy levels accurately – introducing a systematic bias between the cell lines which will not be addressed in a pseudobulk analysis. I am afraid this is potentially a fundamental flaw.

We thank the Reviewer for these important comments. We agree that low mtDNA coverage can limit the ability to detect and accurately measure heteroplasmy. To investigate this, we generated 8x-fold more deeply sequenced data for our HEK293 lines, which improved the raw reads from about 23,000 to 190,000 per cell (10x Genomics recommends up to 25,000 reads) and yielded a substantially increased per-cell mtDNA depth (median: CTRL = 55.6x, KI36 = 154.1x, KIA2 = 255.2x). Importantly, the increased mtDNA depth improved variant detection sensitivity and robustness; however, the relative differences between control and *POLG*^{D274A} knock-in lines remained consistent with our previous results. Likewise, the scmtMMB metrics remained stable across varying sequencing depths (new **Extended Data Fig. 7**), further supporting the robustness of our approach.

As such, we updated our HEK293 results with the more deeply sequenced data in our revised manuscript (**Figs. 1–4**) and further clarified throughout the text whether analyses are performed at the single-cell or pseudobulk level. In addition, we would like to clarify that per-cell mtDNA coverage distributions are shown in **Fig. 1c** (projected onto the UMAP), the total coverage being quite uniform (**Fig. 1e**) and **Extended Data Fig. 1d** (as a violin plot), to which we now also directly refer in the main text and figure legends. We have provided these figure panels also below as **Reviewer Fig. 1** for the Reviewer's more direct consideration.

Reviewer Fig. 1 – Raw mitochondrial DNA (mtDNA) depth in single cells. UMAP projection color-scaled by the mtDNA depth (left), the violin plot of raw mitochondrial DNA depth (middle), and the mtDNA position-wise pseudobulk coverage (right).

In addition, we revised our discussion to emphasize that a minimal, and importantly comparable, mtDNA sequencing depth is required for meaningful biological comparisons. We further point out that biological differences in mtDNA copy number across cell types and analytical methods may need to be accounted for when interpreting such results.

2. The POLG KI line is inducible allowing the authors to turn on and turn off the mutator phenotype. This is very elegant but not capitalised on. Why didn't the authors compare the same cell line before and after a burst of mutagenesis? This would control for many of the confounders they struggle with (eg inherited variants – and critically, the method used to generate the cell line itself which could induced mutations, or lead to rapid segregation of pre-existing mutations through bottleneck effects). I am afraid this could also be another fundamental flaw influencing the main result.

We thank the Reviewer for this thoughtful comment. We would like to clarify that the inducible element in the *POLG*^{D274A} knock-in KIA2 line refers to the tetracycline-inducible mitoEagI restriction enzyme, which cleaves mtDNA but does not affect the proofreading deficiency due to the *POLG*^{D274A} mutation itself. Thus, mitoEagI induction does not generate a burst of mutagenesis but instead results in mtDNA cleavage. However, this is a feature that we did not explicitly utilize in our study. Instead, the *POLG* analysis was conducted with the primary intention of quantifying differences in the mutational burden arising from the *POLG* exonuclease deficiency (as opposed to the kinetics of the system). As such, we used the additional KIA2 line without induction as an additional knock-in sample to benchmark our mutational burden analysis.

To evaluate selection dynamics of mutation under a metabolic bottleneck, we have now included new data where the cell lines were cultured in the presence of glucose or galactose. Notably, the KIA2 line with the higher mutational burden did not tolerate galactose treatment, prohibiting further evaluation. For the KI36 line, about 75% of cells quickly succumbed in the presence of galactose, suggesting a significant bottleneck. Notably, instead of pronounced shifts in heteroplasmy, we observed a statistically significant increase in mtDNA copy number, that we interpret as the primary mode of compensation in this experimental setting. We present these results in new **Fig. 5** and **Extended Data Figs. 8-9**.

However, we believe that different concentrations and/or longer treatments may bring about distinct mutation kinetics. We also note that relatively few confirmed pathogenic mtDNA mutations were present in the lines to begin with, which likely would have exerted more pronounced selective pressures. Also, the mutational burden was relatively uniform across the KI36 line, potentially indicating that the effects of galactose treatment may have also uniformly affected cells as opposed

to selecting for a minor and more tolerant (clonal) population. We believe additional investigations to be necessary and a fruitful avenue to further dissect *POLG* mutator biology, but we would respectfully consider such experiments to be more suitable for future investigation.

We also acknowledge that some residual uncertainty may remain regarding pre-existing mutations, in particular those of low heteroplasmy, and present in only a small proportion of cells. However, such mutations will always be challenging to reliably capture, given detection limits and or sampling bias of all sequencing assays. As such, we would not consider our result to be fundamentally flawed as opposed to being subject to typical experimental and analytical restrictions of this type of data. Analogous to single-cell transcriptomic analysis, the capture of lowly expressed transcripts or small cell populations can be biologically meaningful, but care should be taken with respect to biological interpretation, as we now more explicitly point out limitations in our revised discussion.

Given our primary goal of benchmarking single-cell assessments of mtDNA mutational burden and establishing appropriate quantitative metrics, we believe that our more deeply sequenced HEK dataset, the new galactose experiment, together with the additional revisions to the manuscript, constitute a significant and robust methodological advance for the field.

3. The hash-tagging method is not without difficulties but the authors do not show that their method reliably separates the different cell lines. This is critically important. We need to be confident that the authors are able to reliably resolve single cell sequences to measure the mutation burden in single cells.

We thank the Reviewer for this comment. Hashtag-based multiplexing is a widely used and well-validated approach for single-cell demultiplexing^{1,2}, and we followed these established protocols and analytical workflows. In our dataset, the three HEK293 cell lines showed clear and distinct hashtag signal separation, as illustrated in new **Extended Data Fig. 1c**, demonstrating that the hashtagging method reliably resolved the individual cell lines for confident downstream single-cell mutation burden analyses.

In our revised manuscript, we again applied hashtag-based multiplexing in a more complex setting. In the galactose treatment experiment, we use six hashtag antibodies to distinguish the cell line-treatment combinations (CTRL and KI36 cultured in glucose for 3 days, or galactose for 1 and 3 days). As shown in new **Extended Data Fig. 8b-c**, the six hashtags show well-separated signal distributions with minimal overlap, supporting the robust demultiplexing of all conditions and validating the performance of the methodology as previously benchmarked.

4. Orthogonal validation. The methods they are using are novel and it would be reassuring to see some orthogonal validation eg of the differences in mtDNA content (related to depth), and the presence of linearised mtDNA at high levels. Genuine bulk analysis of the mtDNA, deletion and transcriptomes would provide this.

We thank the Reviewer for this comment. Regarding the linearized mtDNA, we have previously demonstrated its detection via an orthogonal Southern blot, and next-generation sequencing of linker-ligated bulk mtDNA isolated from immunoprecipitated mitochondria³.

Regarding the mtscATAC-seq and variants assay thereof, we emphasize that these have been extensively benchmarked and applied^{2,4-8}. Modified versions have further been adapted by members

of the mitochondrial genetics' community, such as for MitoPerturb-seq⁹. Earlier work demonstrated that single-cell heteroplasmies were further well correlated across different single-cell genomic assays, including scRNA-seq and scATAC-seq^{4,5}. In that work, we also demonstrated that pseudobulk heteroplasmies called from single-cell assays are highly correlated with bulk heteroplasmy measurements from bulk genomic sequencing. Please see, for example, Fig. 2F and Fig. S2E from Ludwig et al.⁴, which we pasted below for the Reviewer's consideration (**Reviewer Fig. 2**).

[editorial note: third party material redacted]

We further conducted downsampling analysis of the *POLG*^{D274A} knock-in line data to assess the detection of mtDNA variants as a function of their allelic frequencies at variable mtDNA sequencing depth. As expected, this is very robust for higher allelic frequency variants (new **Extended Data Fig. 7**). Importantly, also low-heteroplasmy variants (those <1% VAF) were consistently recovered, and their pseudobulk VAF estimates remained highly concordant across variable sequencing depths (**Extended Data Fig. 7f**). Moreover, single-cell mtDNA heteroplasmy estimates were highly consistent, with most per-variant Δ VAF values within $\pm 5\%$ and a global Δ VAF distribution tightly centered at zero when comparing a 4-fold difference in sequencing depth (**Extended Data Fig. 7g**). Together, these analyses suggest that allelic frequency estimates are largely stable once moderate sequencing coverage is achieved.

Further, we note that all HEK293 lines, including the control, would be expected to have the same background rate of potential false-positive mtDNA variants. As can be seen in **Fig. 2d**, the number of additional variants detected in the *POLG*^{D274A} knock-in line is, however, substantially larger than in the control line. As such, it is extremely unlikely that this population and the substantial increase of detected variants within is confounded by false-positive variant calls.

We also note our previously introduced paired-end, strand-aware variant calling workflow, as implemented in the mgatk pipeline, provides higher specificity and more conservative filtering than standard bulk mtDNA workflows⁵. As such, we would be uncertain to what extent bulk analysis can provide orthogonal validation beyond our previously published evaluations⁴. In particular, transcriptomic data are subject to various expression levels and, as such, depths of different mtDNA-encoded genes, with tRNA transcripts not being typically captured (**Extended Data Fig. 12**). Further, we previously identified a significant number of false-positive variant calls, which we speculated is due to RNA polymerase transcriptional error (see **Reviewer Fig. 1b** above, which shows variants detected at the RNA, but not DNA level; some of which being due to RNA editing⁴).

5. It would be extremely reassuring to see the effects of growing the cell lines in galactose, which forces oxidative metabolism. I would anticipate that the selection signal would be much stronger – validating their conclusions experimentally, rather than by inference from single observations in each cell line.

We thank the Reviewer for this comment. As suggested, we cultured the HEK293 control and *POLG*^{D274A} knock-in lines in the presence of glucose or galactose. Notably, galactose treatment was lethal to the KIA2 line, and a large fraction of about 75% of cells from the KI36 line quickly died (new **Extended Data Fig. 8**), suggesting a significant bottleneck. As elaborated above and discussed as part of the Reviewer's prior comment #2 and newly added data in the new **Fig. 5** and **Extended Data Figs. 8-9**, we did not observe pronounced shifts in heteroplasmy but instead a statistically significant

increase in mtDNA copy number, suggesting the primary mode of compensation in this experimental setting.

As discussed above, we believe that varying the concentration or duration of treatment could result in different mutation dynamics. As our cell lines contained relatively few confirmed pathogenic mtDNA mutations, the associated selective pressures were likely limited. Moreover, the relatively uniform mutational burden observed across the KI36 line suggests that galactose treatment may have affected all cells similarly, rather than selecting for a more tolerant subpopulation. While we consider further studies in this direction to be promising, we view them as bestsuited for future independent investigations. However, the significant cell death of the *POLG*^{D274A} knock-in lines provides strong additional evidence that the elevated mutational burden significantly compromised mitochondrial function.

6. What is the provenance of the cell lines? Eg number of passages for the primary and *POLG* KI lines?

For the generation of the *POLG*^{D274A} knock-in lines, single-cell cloning and about two months of culturing were required to expand the populations, during which mtDNA mutations accumulated, followed by cryopreservation of aliquots. Upon thawing, aliquoted cells were cultured for up to three passages (less than two weeks) before single-cell sequencing to minimize the accumulation of additional mtDNA mutations *in vitro*. The passage numbers at the time of sequencing were P18, P19, and P16 for CTRL, KI36, and KIA2, respectively. We have now revised our method section to incorporate this relevant information.

7. I am unclear whether their pipeline excludes the confounding effects of mtDNA sequences embedded in the nucleus (important for the low-heteroplasmy variants)

We thank the Reviewer for this important comment, which we have previously critically evaluated⁵. Since NUMTs are typically not enriched in regions of accessible chromatin, they are not readily amplified by scATAC-seq. In line with this, in the original mtscATAC-seq study by Lareau et al., we have shown that NUMTs only negligibly contribute to single-cell ATAC-seq-based library preparation, expecting only ~1 fragment per cell. This makes NUMT very unlikely to meaningfully confound heteroplasmy estimates. We have pasted the description of our prior evaluation⁵ directly below for the Reviewer's consideration:

“To estimate the number of accessible NUMT fragments that would be assigned to mtDNA, we considered two different approaches. First, we used a public GM12878 dataset from 10X Genomics (<https://www.10xgenomics.com/solutions/single-cell-atac/>) that was aligned to the standard hg19 reference and counted the number of fragments per cell overlapping our NUMT blacklisted regions, which resulted in mean 1.4 and median 1.0 fragments per cell. Second, we used a compendium of DNase accessible peaks from 164 distinct samples from the ENCODE and Roadmap consortia, and estimated that these samples contained a mean 22.6 peaks overlapping our NUMT blacklist. Next, using the GM12878 peak set and the same scATAC-seq dataset, we determined that mean 4.1% of the GM12878 DNase peaks were detected over all cells. The product of these two numbers ($22.6 \times 0.041 = 0.93$ fragments per cell) provides an alternative estimate for the number of accessible chromatin fragments overlapping NUMTs (~1 fragment) that were blacklisted. As our mtscATAC-seq assay generates ~5,000–10,000 mtDNA fragments, we conclude that our blacklist approach yields negligible NUMT contamination.”

8. Definition of de novo variants. As the authors note, there is a threshold for detection of mtSNVs linked to mtDNA coverage. It is therefore impossible to be sure whether any variant is strictly 'de novo' or whether it was present in a small number of cells at low heteroplasmy levels. The authors must temper their conclusions/results with this in mind.

We thank the Reviewer for pointing out this important limitation, which we fully agree with. Along these lines, there may be sampling biases, in particular for variants that are only present in a very small population of cells. As such, in the revised manuscript, we have replaced the term "de novo" with "*POLG*^{D274A}-line specific" to accurately reflect the fact that strictly "de novo" variants cannot be unequivocally defined due to coverage-dependent detection thresholds and other biases, such as sampling. However, we would like to emphasize that, although a strictly "de novo" definition is not possible, we would interpret our deeper sequencing and subsampling analyses to indicate that the sheer number of additional observed mutation events in the *POLG*^{D274A} knock-in cell lines suggests that most detected variants are likely to have occurred "de novo" in this genetic background. We have accordingly revised our presentation of these results, acknowledging the discussed limitations.

9. Related to this – the ability to detect low level heteroplasmies is dependent on the sequencing depth. Their own data shows site-specific differences in coverage. Did the authors account for this in their analysis? Particularly when comparing different mutation classes (eg could it be that pathogenic sites tend to have low coverage, thus limiting their potential for detection). It would be useful for the field to show whether this is an important confounder or not.

We thank the Reviewer for this thoughtful comment. Our data show near-uniform mtDNA coverage across the mitochondrial genome (**Fig. 1e**). This observation is consistent with our previous report⁵, where we demonstrated near-uniform mtDNA coverage with mtscATAC-seq, with minor residual variation attributed to PCR amplification or Tn5 insertion bias. However, in the *POLG*^{D274A} knock-in lines, we observe a relatively lower coverage across positions 0-6000, reflecting the linearization of mtDNA in the *POLG*^{D274A} knock-in background, as previously reported³. To more directly evaluate mtDNA coverage, we plotted the mean coverage stratified by the positions of reported pathogenic variants from MITOMAP, as well as the MLC score interval (**Reviewer Fig. 3** below). These analyses clearly demonstrate that neither potential pathogenic nor highly constrained sites exhibit any tendency toward lower or more variable coverage. As such, we conclude that mutation classes and mtDNA constraint are unlikely to be subject to systematic confounding due to limitations in coverage or mtDNA variant detection.

Reviewer Fig. 3 – Comparable per-base mtDNA mean coverage across pathogenic and constraint variant classes. Violin plots show the per-base mtDNA mean coverage stratified by positions with pathogenic variants annotated in MITOMAP (left) and MLC score intervals [Low (0–0.33), Mid (0.33–0.66), and High (0.66–1)] (right) in control and in *POLG^{D274A}* knock-in HEK293 cell lines.

10. The authors have previously used mtDNA to lineage trace, but they do not do this in the current paper, despite their assertion that there are sub clones within their cell lines. I think this needs greater exploration before they reach the conclusions that they do.

We thank the Reviewer for this comment. In our manuscript, we have exemplarily showcased subclone-specific mtDNA variants within the KI36 line (**Fig. 4e**). Using our previously established pipeline⁵, we further demonstrate the confident assignment of distinct subclonal groups based on shared mtDNA variants (**Reviewer Fig. 4**). While lineage tracing in *POLG^{D274A}* knock-in contexts presents an exciting direction, we believe it has relatively limited added value to the main focus of the current work that is focused on benchmarking and deriving quantitative metrics for single-cell mtDNA mutational burden analyses to evaluate the cellular heterogeneity of mtDNA mutational profiles as demonstrated in our manuscript. Moreover, we note that the higher mutation rate in the investigated setting may require additional benchmarking, as mutational events may arise independently (equivalent to barcode homoplasmy) and thereby confound clonal inference.

Reviewer Figure 4. Subclonal structure of KI36. (a) UMAP of the *POLG^{D274A}* knock-in HEK293 clone KI36, colored by clonotype ID (1-3), identified using the FindClonotype function in the R-package Signac. Manual subcluster labels (C1–C3) are shown; each clonotype has a one-to-one match with a subcluster. (b) Heatmap showing variant allele frequencies of 188 clonally informative mtDNA variants (rows) in individual KI36 cells (columns). Cells are grouped by clonotype (top color bar), highlighting the subclonal structure.

11. Finally, the manuscript refers to phenotypes on several occasions, but their current analysis does not systematically study phenotype. Eg 'these findings clearly suggest POLGD274A phenotypes to be driven by the cumulative mutational burden as compared to the impact of individual or a few deleterious variants as observed in classical mitochondriopathies.' I am afraid this conclusion cannot be drawn from the data presented which does not include these phenotypes and certainly doesn't show a causal role.

We thank the Reviewer for raising this point. We agree that our study does not always directly assess cellular phenotypes, as opposed to introducing and benchmarking a broadly applicable single-cell framework to quantify mtDNA mutational landscapes. In the original manuscript, we showed altered complex I and IV activities of the *POLG*^{D274A} knock-in lines, and our new galactose treatments provide additional evidence of mitochondrial dysfunction attributable to the elevated mtDNA mutational burden (new **Fig. 5** and **Extended Data Figs. 8-9**). The pronounced cell death/ growth inhibition of the *POLG*^{D274A} knock-in lines, elevations in mtDNA copy number in response to galactose, overall stable distribution of single-cell VAFs, and alteration in chromatin accessibility profiles of mitochondrial stress response-related genes in KI36 under galactose treatment provides additional phenotypical evaluation. As such, we have carefully revised the manuscript text to adjust the description of our observations. Our revised version now also acknowledges the need for future investigations and technology development to dissect genotype-phenotype associations and highlights how our framework provides an important foundation for such work.

Reviewer #2 (Remarks to the Author):

In this manuscript by Hsieh et al, the authors leverage a technique mitochondrial single-cell ATAC-seq, which they previously established to investigate heteroplasmy levels of specific variants from patients with mtDNA disease, (Nature Genetics 2022) to measure mtDNA mutational loads and somatic mosaicism. To bench-mark their technique they use HEK-293 cell lines with a knock-in mutation in POLG which causes a proof-reading deficiency of the mtDNA polymerase and elevated mtDNA mutation rates. Interestingly in these lines, the variant allele frequencies (VAFs) of individual cells did not reach classical thresholds for metabolic dysfunction. However, when complex I and complex IV activity was measured in bulk cell cultures, there was metabolic impairment, suggesting multiple mutations can act synergistically to affect mitochondrial function.

The authors then introduce two metrics: single-cell mtDNA mutations per million base pairs (scmtMPM) and heteroplasmy-weighted mitochondrial local constraint scores (scwMSS). These useful measures allow comparative analysis of samples from subjects of different ages and disease groups which is very useful for the research community. They then apply their benchmarked technique and measures to test datasets including control blood samples from young (5 year old) and older (47 year old) controls and 4 patients with the m.3243A>G mutation (aged 29-80). They show that the somatic mtDNA mutation load per cell increases with age in control samples and suggest that the m.3243A>G mutation may affect the accumulation of somatic mtDNA mutations.

Specific comments:

1. The authors have established a protocol to quantify mtDNA mutation load at the single cell level at an unprecedented resolution. Previous single cell work on somatic mtDNA mutations has been at the level of 100s of cells, this was 1000s. The data is very high resolution and supports previous studies which have shown similar mutational spectra and dynamics of age-related somatic mtDNA mutation accumulation.

We thank the Reviewer for the positive and encouraging feedback on the technical advances and resolution achieved by our methodology.

2. This work is of high technical quality and provides a robust framework for assessing mtDNA load/constraints/evolution in other tissue/disease subgroups.

We thank the Reviewer for acknowledging the robustness and broad applicability of our framework for assessing mtDNA mutational load and constraint.

3. I think the title could be clearer, as although the technique is technically multi-omic, other than one experiment towards the end where the nuclear ATACSeq data is used to subset immune-cells, this paper is really focussed on mtDNA sequence analysis. There is very little discussion of the impact (if there is any) of cell type on somatic mtDNA mutagenesis. The majority of the analysis groups all of the different cell types together. I would suggest either fully describing all of the data at the level of cell types, or just using the name of the technique in the title (mitochondrial single-cell ATAC-seq) which will show the focus of the paper.

We thank the Reviewer for this feedback regarding the extent to which we leverage the multi-omic aspect of our dataset. We appreciate the Reviewer's recognition that our primary focus is on high-

throughput single-cell mtDNA genotyping analysis, and the development of a broadly applicable framework for mtDNA mutational landscapes rather than an exhaustive characterization of nuclear scATAC-seq profiles. However, we note, that the scATAC-seq-derived nuclear data is the basis for all cell clustering and cell annotation analyses that is used throughout the manuscript. In addition, in the revised manuscript, we have further expanded the multi-omic analysis specifically i) by comparing chromatin profiles of CTRL vs. *POLG*^{D274A} cells (**Extended Data Fig. 2**), as well as ii) in the new galactose treatment experiment, where we consider the integration of chromatin accessibility and mtDNA genotypes to be most informative (new **Fig. 5** and **Extended Data Fig. 8-9**). Specifically, we demonstrated mtDNA copy number amplification with concomitant chromatin accessibility changes in genes related to mitochondrial stress under galactose-induced metabolic stress in the *POLG*^{D274A} KI36 line. As such, we do consider it appropriate to retain ‘multi-omics’ in the title and hope for the Reviewer’s agreement.

Nevertheless, the investigation of mtDNA mutational-burden–related chromatin accessibility changes in primary human cells represents an important and promising future direction, but we believe that a larger cohort and substantial additional effort will be required to address this question rigorously, which we respectfully consider beyond the scope of the present study and its focus. However, moving forward, we believe our approach provides a framework for diverse future investigations to assess the impact of somatic mtDNA mutagenesis across a variety of biological systems and contexts (e.g., aging and primary mitochondrial disorder), as we now also further outline in our revised **Discussion**.

4. There appears to be three distinct populations of cells in the KI36 cell line, but the data is analysed as one population. Are there specific clonal variants that are driving the clustering in this line?

We thank the Reviewer for this comment. Yes, there are indeed various subclone-specific mtDNA variants that are driving the sub-clustering of the KI36 line, select ones of which we visualized in **Fig. 4e**. Please also consider the heatmap below, which shows 188 mtDNA variants that are shared and/or are specific to each subcluster (see **Reviewer Fig. 4** below). Along these lines, we did note subtle differences in mutational burden as evident by diverging mutational metrics for different gene groups (**Fig. 4c,d**).

Reviewer Figure 4. Subclonal structure of KI36. (a) UMAP of the *POLG*^{D274A} knock-in HEK293 clone KI36, colored by clonotype ID (1-3), identified using the FindClonotype function in the R-package Signac. Manual subcluster labels (C1–C3) are shown; each clonotype has a one-to-one match with a subcluster. (b) Heatmap showing variant allele frequencies of 188 clonally informative mtDNA variants (rows) in individual KI36 cells (columns). Cells are grouped by clonotype (top color bar), highlighting the subclonal structure.

5. Can you be more explicit when describing the confirmed pathogenic mutations on Mitomap? There are a number of different tables on there and I'm guessing you used the 'confirmed' table. It would be good to state this to avoid confusion.

We thank the Reviewer for raising this important point for clarification. In the revised manuscript, we now clarify that the table of confirmed pathogenic variants was obtained from the list of "Confirmed Pathogenic Mutations" in MITOMAP. Variants were further filtered only for those annotated as "Confirmed (Cfrm)", "Confirmed- Pathogenic (Cfrm [P])", or "Confirmed- Likely Pathogenic (Cfrm [LP])". This is now described in the revised **Methods** section, and the list of all pathogenic variants evaluated in our study is now provided as a new **Supplementary Table 4**.

6. For the scmtMPM score, from what I understand, this is normalised to coverage? Can you be explicit that this is the total of the mtDNA aligned reads and not the overall sequencing depth of the ATAC-Seq?

We thank the Reviewer for this question. Yes, the scmtMPM score is indeed normalized to the total number of bases that have been sequenced for the defined mtDNA region (mtDNA-aligned reads), rather than the total ATAC-seq read depth, which would also contain nuclear fragments. We have now added this clarification to our revised Methods section to avoid any ambiguity.

7. I don't really understand how the conclusion was drawn that the 3243 mutation impedes somatic mtDNA mutation accumulation? In figures 5d and 5e both of these patients are relatively young (29 and 35) and the total scmtMPM (-3243A>G) looks to be slightly lower than H47 (closest age-match). However, M60 and M80 are much higher supporting the accumulation of somatic mtDNA mutations with age. What is the evidence that the 3243 variant impedes the accumulation of somatic variants? What happens if you draw similar plots for M60 and M80? I may be misunderstanding the figure, is there an alternative way of presenting this data?

We apologize that this observation has not been sufficiently clearly described. We have deliberately distinguished between the younger (M29 and M35) and older patients (M60 and M80), as at a more advanced age, the pathogenic mt.3243A>G variant has been largely purified away. This is intimated in **Fig. 6b** as the cellular mutational burden significantly drops in the younger patients when excluding the mt.3243A>G variant. In contrast, the burden remains largely unchanged in the older patients. The decreasing levels of the mt.3243A>G variant with age are thereby consistent with prior observations^{7,8,10,11}. This may further suggest that somatic mtDNA mutation dynamics may indeed change as a function of the declining mt.3243A>G variant over the lifetime of the patient. However, we acknowledge the speculative nature of these interpretations.

With respect to the data shown in **Fig. 6d,e**, we have compared how the total scmtMPM changes when including the mt.3243A>G variant (panel d) vs. excluding it (panel e). What we noted is that in cells with a high mt.3243A>G variant allele frequency, the total scmtMPM score was almost entirely driven by the presence of the pathogenic variant, given its removal leads to a significant decrease of the captured mutational burden. As such, we interpreted that cells with a high pathogenic variant burden may be less tolerant to acquiring additional mutations. We also display the data as scatter plots in **Extended Data Fig. 11e** for the older patients (M60 and M80). However, the distribution does not allow for a meaningful smooth kernel density representation, likely due to strong purifying selection

and the resulting sparsity of cells at the high end of the mt.3243A>G VAF distribution. We understand that there is a speculative nature to this interpretation, but we felt this is an intriguing observation revealed through our single-cell somatic mtDNA burden analysis that we wanted to share with the readership. It will be interesting to assess the extent of this phenomenon in additional patients, including those with mitochondriopathies due to other pathogenic mtDNA mutations. We have revised the manuscript accordingly to better clarify the interpretations and future perspectives of the described observation.

Reviewer #2 (Remarks on code availability):

This is not my area of expertise so I did not attempt to run the code.

Reviewer #4 (Remarks to the Author):

The study by Yu-Hsin Hsieh and colleagues presents a study in mitochondrial genomics through the development and application of novel single-cell metrics to quantify mitochondrial DNA (mtDNA) mutational burden and heteroplasmy dynamics. Utilizing mitochondrial single-cell ATAC-seq (mtscATAC-seq), the authors introduce two key metrics—single-cell mtDNA mutations per million base pairs (scmtMPM) and heteroplasmy-weighted mitochondrial local constraint scores (scwMSS)—to capture genome-wide mtDNA variation at single-cell resolution. The study benchmarks these metrics in POLGD274A knock-in cell lines and applies them to primary human immune cells from both healthy donors and patients with mitochondrial encephalomyopathy (MELAS). While the work addresses an important challenge in mitochondrial genetics, offering potentially valuable insights into mitochondrial disease mechanisms, several critical issues and concerns warrant further attention.

We thank the Reviewer for their praise and the recognition of the importance of our work to the field of mitochondrial genetics.

Major Comments:

1. The authors reported 455 variants in control cells versus 2484 and 4651 variants in the KI36 and KIA2 POLGD274A knock-in cells, respectively. Given the substantially higher sequencing depth in the knock-in lines, the increased variant counts may largely reflect depth-related bias. Additionally, the authors should also clarify how uneven mtDNA coverage across the mitochondrial genome was accounted for during variant calling and downstream analyses.

We thank the Reviewer for this important comment. In the revised manuscript, we have now reanalyzed 8x fold more deeply sequenced data from our HEK293 cell lines. This improved the raw reads from about 23,000 to 190,000 per cell (10x Genomics recommends up to 25,000 reads) and yielded a substantially increased per-cell mtDNA depth (median: CTRL = 55.6x, KI36 = 154.1x, KIA2 = 255.2x). This further enabled us to perform downsampling analysis to systematically assess the influence of sequencing depth on mtDNA variant detection (see **Reviewer Fig. 5** below; new **Extended Data Fig. 7**). As one may expect, the total number of detected variants increases with sequencing depth; however, the rate of increase began to plateau at approximately 50x mtDNA coverage, even in the hypermutated *POLG*^{D274A} knock-in background. Because mutation counts are inherently correlated with sequencing depth, we introduced the single-cell mtDNA mutations per million base pairs (scmtMPM) metric, which effectively normalizes for “position-wise” sequencing depth by summing the total number of bases sequenced across the mtDNA regions of interest (i.e., respective gene, or all genes as part of a complex).

Regarding the concern about uneven mtDNA coverage across the mitochondrial genome, our data show near-uniform coverage (**Fig. 1e**), consistent with our previous demonstration and benchmarking of the mtscATAC-seq assay⁵. While some variation in coverage may be attributable to Tn5-related insertion bias and other technical nuances, there are typically no significant differences in coverage patterns between samples.

Variant calling is then performed using the standardized mgatk pipeline, which we have also previously benchmarked in the same study by Lareau et al⁵. This pipeline incorporates a stringent paired-end read strand concordance threshold filter to ensure high-confidence variant detection. Variant allele frequencies (VAFs) are calculated based on position-specific coverage, and only sites supported by at least five reads are retained for downstream analysis. Together, these measures minimize coverage-related artifacts and ensure reliable variant detection across the mitochondrial genome.

Further, we note that differences in mtDNA coverage can also be biological, as different cell types have different mtDNA copy numbers. In the context of the HEK293 lines, we note that the *POLG*^{D274A} knock-in lines indeed have a higher copy number, which has previously been described as a compensatory mechanism^{12–14}.

Reviewer Fig. 5 (Manuscript Extended Data Fig. 7) – Impact of sequencing depth on mtDNA variant detection in *POLGD*^{274A} cell lines. (a) Linear regression of total mtDNA variant detection rate as a function of sequencing depths up to <40M reads. Data points represent the number of total mtDNA variants detected. Regression lines with 95% confidence intervals and equations are shown. (b) LOESS regression of total mtDNA variant counts across the broader range of sequencing depths (10–200M reads). Data points represent the number of total mtDNA variants detected, and LOESS curves with 95% confidence intervals are shown. (c, d) Violin plots of mtDNA variant counts in single cells at (c) shallow sequencing depths (up to 40M reads) and (d) across the broader range (up to 200M). The (c) linear regression and (d) LOESS modeling were performed on the per-depth median number of detected variants. (e) Stream plot of relative heteroplasmy across the various downsampling depths, depicting the overall variant composition. Each band in the stream plot thus represents the relative abundance, by normalizing to the sum summed pseudobulk VAFs of all variants at that depth, of an individual mtDNA variant across downsampling depths. (f) Scatter plots show pseudobulk VAFs of mtDNA variants or all detected variants in KIA2 when comparing datasets downsampled to 50M versus 200M total mtDNA reads. Each point represents a single mtDNA variant. Insets zoom in on variants with VAF <1% to highlight concordance even at low allele frequencies. (g) Density plot of changes in estimated VAF (Δ VAF; 200M vs 50M) for all non-zero-VAF mtDNA variants (7,100 variants) across individual KIA2 cells ($n = 1,894$). (h) Violin plots of scmtMPM and scwMSS in single cells across the fixed subsampled mtDNA sequencing depth (10–200M mtDNA reads).

2. The two metrics (scmtMPM and scwMSS) represent key contributions of this study. However, biases in variant detection and variability in sequencing coverage could substantially influence their

accuracy. The reported median mtDNA sequencing depth, ranging from approximately 10x to 40x, is very low for confident variant calling, particularly for low-frequency variants. Although the authors have implemented measures to minimize these biases, further validation, such as targeted deep sequencing, is needed to confirm the accuracy of variant calls and to robustly support the conclusions drawn from these metrics.

We thank the Reviewer for this comment, which also relates to comments #1 and #4 of Reviewer #1. As suggested by the Reviewer and to assess the impact of sequencing depth on our mtDNA mutational burden metrics, we conducted 8x-fold deeper sequencing of our HEK293 dataset as also elaborated on in our response to the Reviewer's prior comment. This enabled us to perform downsampling analyses to simulate various sequencing depths across the different HEK293 cell lines. Importantly, both scmtMPM and scwMSS metrics remained stable across different downsampling depths (new **Extended Data Fig. 7h**), demonstrating that our approach effectively mitigates depth-related bias in mutational burden quantification. Accordingly, we updated our HEK293 results with the more deeply sequenced data in our revised manuscript (**Figs. 1–4**).

We further conducted downsampling analysis of the *POLG*^{D274A} knock-in line data to assess the detection of mtDNA variants as a function of their allelic frequencies at variable mtDNA sequencing depth. As expected, this is very robust for higher allelic frequency variants, as shown in new **Extended Data Fig. 7**. Notably, low-heteroplasmy variants (those <1% VAF) were also consistently recovered, and their pseudobulk VAF estimates remained highly concordant across depths (**Extended Data Fig. 7f**). Moreover, single-cell mtDNA heteroplasmy estimates were highly consistent, with most per-variant Δ VAF values within $\pm 5\%$ and a global Δ VAF distribution tightly centered at zero when comparing up to 4-fold differences in sequencing depth (**Extended Data Fig. 7g**). Together, these analyses suggest that allelic frequency estimates are largely stable once moderate coverage is achieved.

Further, we note that all HEK293 lines, including the control, would be expected to have the same background rate of potential false-positive mtDNA variants. As can be seen in **Fig. 1d**, the number of additional variants detected in the *POLG*^{D274A} knock-in line is, however, substantially larger than in the control line. As such, it is extremely unlikely that this population and the large increase of detected variants within is confounded by false-positive variant calls.

Regarding the mtscATAC-seq and variants assay thereof, we emphasize that these have further been extensively benchmarked and applied^{2,4–8}. Modified versions have further been adapted by members of the mitochondrial genetics community, such as for MitoPerturb-seq⁹. Earlier work further demonstrated that single-cell heteroplasmy estimates were well correlated across different single-cell genomic assays, including scRNA-seq and scATAC-seq^{4,5}. In that work, we also demonstrated that pseudobulk heteroplasmy estimates called from single-cell assays are highly correlated with bulk heteroplasmy measurements from bulk genomic sequencing. Please see, for example, Fig. 2F and Fig. S2E from Ludwig et al., which we pasted below for the Reviewer's consideration (**Reviewer Fig. 2a,b**).

[editorial note: third party material redacted]

We also note our previously introduced paired-end, strand-aware variant calling workflow, as implemented in the mgatk pipeline, provides higher specificity and more conservative filtering than standard bulk mtDNA workflows⁵. As such, we would be uncertain to what extent bulk analysis, including via targeting of select variants among the pool of >10,000 detected variants, and challenges

around sampling bias of very low frequency variants, can provide full orthogonal validation beyond our previously published evaluations^{2,4-8}. However, we acknowledge the residual uncertainty surrounding very low allele frequency heteroplasmic variants that may be only present in a small number of mtDNA copies or cells but would suggest that such variants are unlikely to meaningfully add to a mutational burden analysis and/or be biologically impactful. We now discuss this in our revised discussion, including the need for continued refinement of experimental and computational methods, such as the integration of unique molecular identifiers and duplex-sequencing approaches¹⁵.

3. The analysis assumes that variants unique to POLGD274A lines are *de novo*, but clonal expansions or bottlenecks could also influence variant distributions. Could the authors elaborate on how these factors were considered or controlled for in their analyses?

We thank the Reviewer for highlighting this important aspect of mitochondrial genetics. We agree that strictly "*de novo*" variants cannot be unequivocally established given the features of mtDNA inheritance (e.g., clonal expansion, bottlenecks) and technical factors (e.g., sampling bias, sequencing depth). Thus, we have revised our manuscript by replacing the term "*de novo*" with "*POLG^{D274A}*-line specific" to reflect this concern accordingly. To mitigate potential confounding effects from clonal expansion and bottleneck effects, our downstream analyses only considered variants absent from the parental HEK293 control cell line.

With respect to clonal expansions and bottleneck effects, these were not explicitly considered, as we would be uncertain to what extent these may present potential confounders. Specifically, clonal expansions would likely increase the VAF of the mtDNA variants and thus enhance their detectability. For example, **Fig. 4e** shows a number of subclones specific to the line KI36. Given their recurrent detection across a number of cells and their concentration in specific subclusters, one can consider these very high-confidence variant calls.

Further, our read-downsampling analysis suggests the observed mutation events to be significantly more prominent in the *POLG^{D274A}* knock-in cell lines (**Extended Data Fig. 7a-d**), along with a lower VAF distribution (**Fig. 2f,g**). Together, this suggests that the variants are indeed likely to have occurred "*de novo*" in this genetic background. However, we agree that some residual uncertainty remains related to the confident detection of very low frequency mutations, and as such have reworded the description of these results and point out respective limitations in our revised manuscript. Please see also our response to comment #2 regarding the background false-positive rate.

With respect to bottleneck effects, we would believe that longitudinal sampling experiments with a larger population cohort may be more explicitly tailored to answer such questions, which will require more comprehensive efforts, which we would respectfully consider to be more suitable for future investigations. Nonetheless, in our revised manuscript and as suggested by Reviewer #1, we have now conducted a more defined pilot experiment in *POLG^{D274A}* knock-in cell lines, where we utilized galactose to create a metabolic bottleneck by enforcing OXPHOS metabolism (new **Fig. 5**). Notably, we were able to detect various minor shifts in the cumulative heteroplasmy at the single cell level but did not observe pronounced shifts in heteroplasmy for most variants in KI36 cells (**Fig. 5e**). Interestingly, a compensatory increase in mtDNA copy number was observed upon galactose treatment, demonstrating that our approach is able to capture diverse nuances of mitochondrial genetics, including small heteroplasmy shifts.

4. The authors incorporate a mitochondrial genome constraint model based on population genetics data from gnomAD. Have the authors considered the impact of population diversity, mitochondrial haplotypes, as well as the cell-type-specific or context-dependent functional constraints?

We thank the Reviewer for raising this important question of human mitochondrial genetics. In our study, the mutational burden analyses were restricted to heteroplasmic variants. Haplogroup-associated variants, which are typically present at homoplasmy, were excluded from the burden analyses. In theory, as these variants are also typically invariant across the population if sampled from the same donor, these would also not necessarily present a confounding element to the burden analysis.

Moreover, in the original work describing the constraint model¹⁶, the authors did benchmark their MLC score framework against large-scale population data (i.e., MITOMAP, ClinVar, HelixMTdb) to demonstrate that associations between MLC-derived MSS and human phenotypes remained significant with or without adjusting for the haplogroup.

With respect to cell-type-specific or context-dependent functional constraints, we believe that no appropriate data would be available to definitely answer this question. The constraint model is ultimately based on human population data that is most powered to assess primarily germline genetic variation that is well captured at the level of human populations, but not at the level of individual cell types. Potentially, the Reviewer is alluding to the often cell-type-specific phenotypes observed in patients with mitochondriopathies. For example, our prior work demonstrated that pathogenic mtDNA variants/deletions are not well tolerated in human subset of T cells^{7,8}, which may be analogous to the cell-type specific effects the Reviewer may be referring to. However, to fully account for cell-type and context, we are likely to require the cell-/organ- and context-specific evaluation of the impact of variants, which would present a substantial effort. While invaluable, we believe this is clearly beyond the current scope of our manuscript.

5. Have the authors explored integrating other variant annotations, like predicted pathogenicity scores or conservation, to help refine the MLC-based scoring?

We thank the Reviewer for this comment. Here, we would kindly refer the Reviewer to the original article by Lake et al.¹⁶, where the MLC framework was systematically validated against population-level databases, including MITOMAP, ClinVar, and HelixMTdb. The study thereby demonstrated that MLC scores exhibit strong concordance with known pathogenicity and evolutionary conservation. Additionally, the authors also provided experimental validation of the MLC score using base-editing approaches, further supporting its robustness. While we could envision further refinement of the MLC framework (e.g., inclusion of larger populations and of more diverse ancestry), we believe that our current implementation already provides a strong genetic basis for our single-cell evaluations of mtDNA mutational burden. However, of course, the presented scWMSS framework is adaptable and could readily integrate future refinements surrounding MLC scoring and potentially other future relevant models, that may, for example, more comprehensively integrate data across diverse human populations and more advanced variant annotation.

6. Given that mtscATAC-seq simultaneously captures mtDNA variants and chromatin accessibility, have the authors investigated potential interactions between mtDNA mutational burden and nuclear chromatin states? Such as, whether/how nuclear-mitochondrial interactions influencing variant

dynamics? whether cells with higher mutational burden exhibit altered chromatin accessibility; Additionally, single-cell Multiome technique, which profile both transcriptomes and chromatin accessibility in individual cells, could provide an even more powerful platform to investigate the functional consequences of mtDNA mutations. Have the authors considered applying or integrating such approaches?

We thank the Reviewer for this important comment. In our revised manuscript, we now more explicitly leverage the chromatin accessibility modality, to investigate nuclear gene regulatory responses to an elevated mtDNA mutational burden and galactose-induced metabolic bottlenecks. Specifically, we performed a differentially accessible gene (DAG) analysis comparing **i**) *POLG*^{D274A} knock-in vs. CTRL HEK293 cells (new **Extended Data Fig. 2**), and **ii**) *POLG*^{D274A} knock-in KI36 line under glucose vs. galactose conditions (new **Fig. 5**). The first analysis revealed profound chromatin remodeling in both KI36 and KIA2 lines. Shared upregulated and downregulated DAGs were enriched for genes involved in extracellular matrix remodeling (e.g., protocadherins gene family, *PCDHs*), transcriptional regulation (e.g., zinc-finger transcription factors, *ZNFs*), mtDNA maintenance (e.g., *MGME1*), and respiratory chain assembly, indicative of broad nuclear epigenetic responses to an elevated mtDNA mutational burden. Similarly, DAG analysis comparing glucose- vs. galactose-treated *POLG*^{D274A} cells (KI36 line) identified genes involved in mitochondrial/ER stress and inflammatory signaling, indicative of epigenetic rewiring and adaptation in the presence of a profound metabolic bottleneck. Together, these findings showcase the capability of mtscATAC-seq to capture biologically meaningful nuclear-mitochondrial interactions linked to mitochondrial dysfunction.

With respect to the Reviewer's suggestion regarding the utility of the single-cell Multiome approach, we agree that this approach is a powerful platform to investigate the functional consequences of mtDNA mutations and is highly complementary to our current mtscATAC-seq framework. Along these lines, our prior work identified extensive gene regulatory changes in human primary immune cells with pathogenic mtDNA deletions revealed via scRNA-seq⁸. However, the number of samples were still limited. To meaningfully extend this line of work, we are aiming to profile larger cohorts of healthy donors as well as mitochondrialopathy patients to be sufficiently well-powered and account for nuclear as well as mitochondrial genetic heterogeneity. Here, and for the Reviewer's consideration, we have included some of our preliminary work (see **Reviewer Fig. 6** below), demonstrating the potential for future applications of our MMB metrics.

Specifically, we stratified cells with a very low vs high mutational burden in a select healthy donor and a MELAS patient (**Reviewer Fig. 6a,c**), and performed differential transcriptome analysis in a cell type-specific manner. While a substantial number of differentially expressed genes were identified in both the healthy donor and the MELAS patient cells in various cell types, the results yet lacked statistical significance (**Reviewer Fig. 6b,d**). Additionally, in the MELAS context, the differences could potentially be dominated by the strongly pathogenic mt.3243A>G variant. While the investigation of mtDNA mutational-burden-related chromatin accessibility and transcriptome changes in primary human blood cells represents an important and promising future direction, we believe that for these types of analysis, one would require profiling of a significantly larger number of cells and donors, analogous to eQTL (expression quantitative trait locus) studies, which aim to identify how genetic variations influence gene expression levels by mapping genetic variants to gene expression phenotypes¹⁷⁻¹⁹. However, we hope the Reviewer agrees with our reasoning, that these would readily present independent and stand-alone efforts. Importantly, and moving forward, we believe our approach provides a framework for a variety of future investigations to assess the impact of somatic mtDNA mutagenesis across a variety of biological systems and contexts (e.g., aging and primary mitochondrial disorders).

Reviewer Fig. 6 Preliminary integrated analysis of transcriptomic profiles and mtDNA mutational burden in a healthy donor and a MELAS patient PBMC datasets. (a) Joint transcriptome–chromatin accessibility co-embedded weighted-nearest-neighbor (WNN) UMAP projection of PBMCs from a healthy donor, colored by major immune cell types (left) and scmtMPM (right). (b) Violin plots of selected differentially expressed transcripts comparing high-scmtMPM cells (red; scmtMPM > 65th percentile) and low-scmtMPM cells (blue; scmtMPM < 35th percentile), stratified by cell type. (c) WNN-UMAP projection of PBMCs from a MELAS patient, colored by major immune cell types (left) and scmtMPM (middle), and the VAF of the pathogenic variant mt.3243A>G. (d) Violin plots of selected differentially expressed transcripts in the MELAS patient, using the same high- vs low-scmtMPM comparison as in (b). Differential expression analysis was performed using the `FindMarkers` function in R-package Seurat (v4.0.3) using the default Wilcoxon Rank Sum test.

7. It would strengthen the impact of the study if the authors included functional assays to see how mtDNA variants actually affect cells. Further linking the scmtMPM and scwMSS metrics to cellular phenotypes would add important biological context to the current study.

We thank the Reviewer for this important suggestion. As also suggested by Reviewer 1 (comment #5), we have now included new data where the *POLG*^{D274A} knock-in cell lines were cultured in the presence of galactose, enforcing OXPHOS utilization. This enabled us to evaluate potential selection dynamics of cells and mutations under a metabolic bottleneck, thereby providing an additional layer of phenotypic characterization. We present these results in new **Fig. 5** and **Extended Data Figs. 8-9**. Specifically, the *POLG*^{D274A} knock-in cell lines exhibited a strong proliferation defect in galactose-containing medium, indicating a pronounced mitochondrial defect. Notably, instead of pronounced shifts in heteroplasmy, we observed mtDNA amplification with concomitant chromatin accessibility changes in genes related to mitochondrial stress, suggesting the primary modes of compensation in this experimental setting.

However, as we now note in the revised **Discussion**, we acknowledge that varying the concentration or duration of treatment could result in different mutation dynamics. Moreover, the relatively uniform mutational burden observed across the KI36 line suggests that galactose treatment may have affected all cells similarly, rather than selecting for a more tolerant (clonal) subpopulation. While we consider further studies in this direction to be promising, we respectfully consider them as best suited for future investigations outside of the scope of this manuscript. Nevertheless, the decreased complex I and IV activities (**Extended Data Fig. 1a**) and substantial galactose-induced cell death/growth defects observed in the *POLG*^{D274A} knock-in cell lines (**Extended Data Fig. 8a**) provides strong additional evidence that the elevated mutational burden significantly compromises mitochondrial function.

Moving forward, and as we now mention in our revised **Discussion**, the direct integration of alternative phenotyping approaches, such as single-cell metabolite profiling²⁰ in conjunction with scmtMPM/scwMSS measurements are likely to provide a more direct evaluation on how mtDNA mutational burden affects cells. However, we are currently unaware of an approach that would enable ready integration of the detection of both modalities. Future efforts will be required to catalyze such multi-omic analyses beyond the level of classical genomic approaches.

8. Could the authors expand on the translational implications of their findings. In particular, discussing how feasible it might be to apply the scmtMPM and scwMSS metrics in clinical or therapeutic settings, along with any potential challenges, would add valuable perspective.

We thank the Reviewer for this valuable suggestion. In the revised manuscript, we have now added a new paragraph in the **Discussion** outlining the potential translational applications of our scmtMMB metrics. Specifically, we highlight recent discoveries linking mtDNA mutations and heteroplasmy to clinical outcomes in oncology and hematological malignancies. Moreover, elevations in mutational levels have been associated with aging phenotypes in mutator mice²¹⁻²³. As such, our single-cell framework is expected to be broadly applicable across different biological contexts, though appropriate cohort sizes will be necessary to be adequately powered to detect respective associations. Along these lines, base-editing approaches are likely to catalyze the experimental validation of such observations.

9. The manuscript currently lacks enough detail about the statistical analyses used. The authors should include clear descriptions of the statistical methods applied throughout the study to better support their findings and improve transparency.

We thank the Reviewer for this important comment. We have accordingly revisited our descriptions of the applied methodology and statistical approaches to add more detail and clarity.

Other Comments:

It would be helpful if the authors provided a summary table listing all variants detected in this study, including average sequencing depth, the number of cells carrying each mutation within each group, and other relevant information.

We thank the Reviewer for this suggestion. We now provide a comprehensive **Supplementary Table 1** listing all high-confidence variants, coverage, cell counts, heteroplasmy, and annotations.

References

1. Stoeckius, M. *et al.* Simultaneous epitope and transcriptome measurement in single cells. *Nat Methods* **14**, 865–868 (2017).
2. Mimitou, E. P. *et al.* Scalable, multimodal profiling of chromatin accessibility, gene expression and protein levels in single cells. *Nat Biotechnol* **39**, 1246–1258 (2021).
3. Peeva, V. *et al.* Linear mitochondrial DNA is rapidly degraded by components of the replication machinery. *Nat Commun* **9**, 1727 (2018).
4. Ludwig, L. S. *et al.* Lineage Tracing in Humans Enabled by Mitochondrial Mutations and Single-Cell Genomics. *Cell* **176**, 1325–1339.e22 (2019).
5. Lareau, C. A. *et al.* Massively parallel single-cell mitochondrial DNA genotyping and chromatin profiling. *Nat Biotechnol* **39**, 451–461 (2021).
6. Lareau, C. A. *et al.* Mitochondrial single-cell ATAC-seq for high-throughput multi-omic detection of mitochondrial genotypes and chromatin accessibility. *Nat Protoc* **18**, 1416–1440 (2023).
7. Walker, M. A. *et al.* Purifying Selection against Pathogenic Mitochondrial DNA in Human T Cells. *N Engl J Med* **383**, 1556–1563 (2020).
8. Lareau, C. A. *et al.* Single-cell multi-omics of mitochondrial DNA disorders reveals dynamics of purifying selection across human immune cells. *Nat Genet* **55**, 1198–1209 (2023).
9. Burr, S. P. *et al.* MitoPerturb-Seq identifies common and gene-specific single-cell responses to mitochondrial DNA depletion and heteroplasmy. *bioRxiv* doi:10.1101/2025.07.08.663208.
10. Franklin, I. G. *et al.* T cell differentiation drives the negative selection of pathogenic mitochondrial DNA variants. *Life Sci Alliance* **6**, (2023).
11. Zhang, J. *et al.* Antigen receptor stimulation induces purifying selection against pathogenic mitochondrial tRNA mutations. *JCI Insight* **8**, (2023).
12. Hedberg-Oldfors, C. *et al.* Deep sequencing of mitochondrial DNA and characterization of a novel mutation in a patient with arPEO. *Neurol Genet* **6**, e391 (2020).
13. Salminen, T. S. *et al.* Mitochondrial genotype modulates mtDNA copy number and organismal phenotype in *Drosophila*. *Mitochondrion* **34**, 75–83 (2017).
14. Filograna, R. *et al.* Modulation of mtDNA copy number ameliorates the pathological consequences of a heteroplasmic mtDNA mutation in the mouse. *Sci Adv* **5**, eaav9824 (2019).
15. Lawson, A. R. J. *et al.* Somatic mutation and selection at population scale. *Nature* **647**, 411–420 (2025).
16. Lake, N. J. *et al.* Quantifying constraint in the human mitochondrial genome. *Nature* **635**, 390–397 (2024).
17. Nathan, A. *et al.* Single-cell eQTL models reveal dynamic T cell state dependence of disease loci. *Nature* **606**, 120–128 (2022).
18. Zhang, Z. *et al.* Unveiling genetic signatures of immune response in immune-related diseases through single-cell eQTL analysis across diverse conditions. *Nature Communications* **16**, 7134 (2025).

19. Hong, S. E. *et al.* Single-cell eQTL analysis identifies genetic variation underlying metabolic dysfunction-associated steatohepatitis. *Nature Genetics* **57**, 1638–1648 (2025).
20. Delafiori, J. *et al.* HT SpaceM: A high-throughput and reproducible method for small-molecule single-cell metabolomics. *Cell* **188**, 6028–6043.e11 (2025).
21. Trifunovic, A. *et al.* Premature ageing in mice expressing defective mitochondrial DNA polymerase. *Nature* **429**, 417–423 (2004).
22. Kujoth, G. C. *et al.* Mitochondrial DNA mutations, oxidative stress, and apoptosis in mammalian aging. *Science* **309**, 481–484 (2005).
23. Maclaine, K. D., Stebbings, K. A., Llano, D. A. & Havird, J. C. The mtDNA mutation spectrum in the PolG mutator mouse reveals germline and somatic selection. *BMC Genom Data* **22**, 52 (2021).

Response to Reviewers

We thank the Reviewers for their positive feedback, which further helped to improve the manuscript. We provide a detailed point-by-point response to all comments in blue and have modified the manuscript based on the valuable suggestions and journal formatting guidelines, including shortening of the main text.

Detailed Response to Reviewers

Reviewer #1 (Remarks to the Author):

I thank the authors for their thorough and meticulous response to my first review, and particularly for the generation of new data. This, of course, raises further questions which they may want to consider.

1. The authors generated more reads from the three cell lines. Does the increased read depth influence their results and conclusions?

We thank the Reviewer for this follow-up question. The increased sequencing depth improved the sensitivity and robustness of mtDNA variant detection, particularly for low-frequency variants. However, it did not change the overall results or conclusions of our study. While deeper sequencing detected additional variants, the relative differences between CTRL and *POLG^{D274A}* knock-in lines remained comparable, and the distributions of scmtMPM and scwMSS were stable across a wide range of sequencing depths, as illustrated in our downsampling analyses (**Supplementary Fig. 7**).

2. Why is the read depth different between the cell lines? I presume this reflects a difference in mtDNA content. Has this been validated using an orthogonal method?

We agree that the observed differences in read depth likely reflect biological differences in mtDNA content and the accumulation of linearized mtDNA, as previously described using an orthogonal method such as Southern blot in Peeva et al. (2018, PMID: 29712893) and Trombly et al. (2023, PMID: 37237953). More broadly, our prior work has shown that the percentage of recovered mitochondrial reads varies across tissues, as revealed by our reanalysis of Genotype-Tissue Expression (GTEx) data. For example, this analysis showed particularly high mitochondrial read percentages and coverage in human muscle, brain, and heart compared with other tissues, which aligns well with their higher mitochondrial content. Please see Figure 4a-c from DOI: [10.1016/j.cell.2019.01.022](https://doi.org/10.1016/j.cell.2019.01.022), which we have also copied here for the Reviewer's consideration.

[editorial note: third party material redacted]

3. The comparison of heteroplasmy value using pseudo and real-bulk analysis really misses the point I was making – that low coverage compromises the ability to call and measure heteroplasmy reliably. I do not think the current data really addresses this.

We thank the Reviewer for raising this concern again. The reliable detection of very low-frequency variants is generally challenging in genomics, whether in nuclear or mitochondrial genome sequencing. We believe this may ultimately be addressable only via very high-fidelity sequencing platforms, such as NanoSeq, a duplex sequencing method with an error rate lower than five errors per billion base pairs, as introduced by Lawson et al. (2025, PMID: 41062696). However, these technical principles have yet to be adopted for single-cell sequencing. However, as explained in the prior round of review, the use of strand-concordance already provides a quite conservative threshold for variant calling in our single-cell data. Nevertheless, we acknowledge in the revised discussion that the detection of very low-frequency variants must be interpreted with caution, both technically and biologically, with respect to their functional impact.

4. The authors should reword their section on the 'metabolic bottleneck'. As written, it conflates two issues: (1) a bottleneck – if present you should see heteroplasmy levels changing by accelerated random drift – or the random loss/gain of heteroplasmies; (2) the metabolic stress which they suggest has killed a large proportion of cells. If this is the case, then you would expect less pathogenic variants after the metabolic challenge. It is not clear to me whether the authors saw this or not.

We thank the Reviewer for this important conceptual clarification and agree that the original wording conflated two distinct processes. We have accordingly revised the manuscript for clarity. As demonstrated in our original submission, pathogenic variants were less prevalent than other variant types, which we interpreted as cells with pathogenic variants being selected against over the course of culture, independent of galactose treatment. As such,

we believe a different experimental setup is likely necessary to reaffirm the Reviewer's notion, as presented in the discussion of our work.

Reviewer #2 (Remarks to the Author):

I enjoyed reading the revised version of this manuscript. I am happy that that authors have addressed my comments and the additional analyses have strengthened the manuscript.

We sincerely thank the Reviewer for this praise and the valuable feedback during the review of our work.

Reviewer #3 (Remarks to the Author):

I thank the authors for their thorough responses. My previous concerns have been addressed, and I have no further comments on the manuscript.

We are grateful to the Reviewer for their time and suggestions to improve our manuscript.